# Debris Flow Scale Prediction Based on Correlation Analysis and Improved Support Vector Machine

Li Li [1,*], Zhongxu Zhang [1], Dongsheng Zhao [2], Yue Qiang [1], Bo Ni [1], Hengbin Wu [1], Shengchao Hu [1] and Hanjie Lin [1]

1   Civil Engineering College, Chongqing Three Gorges University, Wanzhou, Chongqing 404100, China; zzx13983468130@163.com (Z.Z.); qiangyue-320@163.com (Y.Q.); boni202003@163.com (B.N.); hbw8456@163.com (H.W.); hushengchao1999@163.com (S.H.); 15730552177@163.com (H.L.)
2   Architecture and Engineering College, Sichuan Institute of Industrial Technology, Deyang 618500, China; zhaodongsheng1996@163.com
*   Correspondence: 20120011@sanxiau.edu.cn

**Abstract:** The occurrence of debris flows are a significant threat to human lives and property. Estimating the debris flow scale is a crucial parameter for assessing disaster losses in such events. Currently, the commonly used method for estimating debris flow runoff relies on fitting techniques, which often yield low prediction accuracy and limited data representation capabilities. Addressing these challenges, this study proposes an improved grey wolf algorithm optimized support vector machine prediction model. The model's effectiveness is validated using data from 72 debris flow events in Beichuan County. The results demonstrate a prediction accuracy of 95.9% using this approach, indicating its strong predictive capabilities for debris flow scale. Additionally, it is observed that the basin area, the basin relative, and the main channel length are the key factors influencing debris flow scale in Beichuan County.

**Keywords:** correlation analysis; debris flow scale; grey wolf optimizer algorithm; support vector machines

## 1. Introduction

The scale of debris flow refers to the amount of loose solid material flushed out by the debris flow from its formation to movement. Usually, the scale of debris flows is defined as the volume size that eventually leads to their formation [1–3]. It can be divided into four categories according to the size of the debris flow:

1.  Small debris flow refers to the amount of loose solid material flushed out if less than 10,000 cubic meters.
2.  Medium-sized debris flow refers to the volume of loose solid materials flushed out between 10,000 cubic meters and 100,000 cubic meters.
3.  Large debris flow refers to the volume of loose solid materials flushed out between 100,000 cubic meters and 1 million cubic meters.
4.  Giant debris flow refers to the amount of loose solid material washed out if more than 1 million cubic meters.

Generally speaking, the larger the debris flow, the more serious the disaster, which may cause greater damage and losses to human society and the natural environment. Therefore, accurate assessment and prediction of the scale of debris flow is crucial to taking effective disaster prevention and mitigation measures.

De Haas et al. [4] designed a debris flow volume prediction model based on the area of debris flow accumulation fan. However, their research did not examine the impact of lithology and climate on the debris flow volume in the study area. The estimation of debris flow volume based on fan area and the lack of factors may lead to errors in the estimation of debris flow volume. Ma et al. [5] used only loose body volumes to establish a mathematical

and statistical relationship with debris flow volumes. Although the correlation coefficient is as high as 0.928, other factors may also be significantly related to the prediction of debris flow. Gartner et al. [6] used multiple linear regression to predict debris flow volumes at seven different sites. They chose different influencing factors at each location. However, a linear regression model for one study site may not be suitable for another. Consequently, numerous studies are necessary for estimating mudslide volumes in other areas, which is an arduous task. Chang et al. [7] identified factors that affect debris flow volume, including watershed area, landslide area, stream length, average stream and watershed slope, form factor, and geological index. They constructed an empirical formula model, with significant results, particularly when applied to areas with heavy rainfall. However, it was demonstrated that predicting debris flow volume is a challenging undertaking, and any empirical model should be augmented with additional approaches. Upon analysis of the empirical formulae, it becomes apparent that the prediction models for debris flow sizes are statistically based and acquired through mathematical fitting. The formula selection of these prediction models is highly subjective. This leads to the selection of different functions which have a great influence on the fitting results, and the data representation is insufficient. Furthermore, considering the regional characteristics of mudslides, the factors influencing the size of mudslides vary greatly across different regions. Therefore, the same formula cannot be applied uniformly across all regions without considering their unique features and conditions.

With the development of machine learning, more and more intelligent algorithms are used in disaster prevention and prediction [8]. Tang and colleagues [9] utilized an artificial neural network (ANN) to predict the volume of debris flow, and the results showed that the model could achieve an accuracy of 78.33%. In addition, they mentioned that the more samples and meteorological data contained in the data set, the higher the prediction accuracy. They also found that the prediction accuracy of mesoscale and large-scale debris flow is higher than that of small scale. Lee et al. [10] also used an ANN to predict the volume of debris flow under extreme weather in Korea and compared the model with three regression equations. Their model results had an $R^2$ value of 0.822 and an MSE value of 0.022. The three regression equations had $R^2$ values of 0.703, 0.703, and 0.691, respectively, and none of them fitted as well as the ANN model. This also verifies our analysis that the same equations are not characterized by regional expansion. Huang et al. [11] employed an adaptive Boost machine learning algorithm that integrates extreme learning machine and particle swarm optimization to forecast the volume of debris flow. The model demonstrates high statistical validity and accuracy, yielding a MAPE of less than 0.1. The validation of their model in other study areas also produced MAPE results ranging between 0.11 and 0.16. Above examples of the application of the machine learning algorithms show that they can overcome the limitations of empirical formulae. Therefore, it highlights their great potential in practical applications.

Support vector machine is an algorithm based on a statistical theory proposed to minimize structural risks [12]. Intelligent algorithms have better promotion capabilities and can overcome the shortcomings of traditional statistical learning theories. Therefore, researchers widely use them in the recognition and classification texts, as well as medical and health, vehicle traffic, failure mode recognition, and other fields [13–16]. Nonetheless, the more rational the internal parameters of the SVM model are, the better the performance of the model [17].

Therefore, the implementation of optimization algorithms is essential for SVM optimization. Swarm Intelligence algorithms, also known as SI algorithms, are commonly used. These algorithms have demonstrated remarkable efficiency in solving complex problems within reasonable timeframes [18]. The Grey Wolf Optimization algorithm (GWO) is a novel SI algorithm. It is inspired by the hierarchical structure and hunting behavior of grey wolves and aims to identify the optimal solution [19]. In studies predicting landslide displacement, GWO is employed to identify optimal parameters for the ELM algorithm. The results show that the GWO-ELM model has superior generalization capability and higher

prediction accuracy. In the shale gas geosteering discriminant model, GWO was utilized to identify the globally optimal parameters in SVM. The GWO-SVM model has a significant improvement in the average crossover rate and prediction accuracy. Compared with the original model, it increased by 5.38% and 7.74%, respectively [20,21]. In addition, the GWO exhibits exceptional competitiveness when compared to other optimization algorithms, such as PSO, GSA, DE, EP, and ES [19].

The aforementioned examples highlight the considerable benefits of the grey wolf algorithm in determining global parameters. Thus, the accuracy of the initial algorithm and the generalization ability of the model are improved. Consequently, this study introduces the grey wolf algorithm to identify the internal parameters of the SVM model and ameliorate its performance.

However, during the final stage of the GWO algorithm's operation, all grey wolves within the population converge towards α wolves, which denote the optimal solution. This ultimately results in a loss of population diversity, local convergence, and premature algorithm convergence. It has been demonstrated that Levy flights effectively locate desirable solutions through random search. This paper presents Levy flights to optimize the Grey Wolf algorithm and address the issue of local and premature convergence in the algorithm's later stages [22]. Inspired by this, using Levy flights to optimize it, an improved GWO algorithm (IGWO) is proposed. Then IGWO is utilized to optimize internal parameters of the SVM algorithm, resulting in an improved performance of the SVM algorithm. Finally, a debris flow volume prediction model is established based on the improved IGWO-SVM algorithm.

In this study, three input factors are selected using correlation analysis. These factors are then fed into the IGWO-SVR algorithm to predict the volume of 72 mudslides in Beichuan County. Section 2 details the correlation analysis, the model construction process, and the comparison of models. Section 3 presents the study area and model results. Lastly, Section 4 thoroughly discusses the contents of this paper and future work that needs further improvement. Section 5 presents the conclusions of the IGWO-SVR prediction model used in the study area of this paper.

## 2. Method

### 2.1. Spearman Correlation Analysis

Spearman correlation coefficient [23] is also called rank correlation coefficient or order correlation coefficient. It uses the rank of two variables for linear correlation analysis to measure whether the two variables are monotonically correlated. The correlation coefficient $\rho$ of this method is defined as the Spearman correlation coefficient between the ranks of two n-dimensional random variables X = (X1, X2, X3... Xn) and Y = (Y1, Y2, Y3... Yn).

$$p = \frac{\sum\limits_{i=1}^{n}(r_i - \bar{r})(s_i - \bar{s})}{\sqrt{\sum\limits_{i=1}^{n}(r_i - \bar{r})^2}\sqrt{\sum\limits_{i=1}^{n}(s_i - \bar{s})^2}} \tag{1}$$

In equations, $r_i$ and $s_i$ correspond to the ranks of $x_i$ and $y_i$, respectively, for $i = 1, 2..., n$. The value of $\rho$ falls within the range of $[-1,1]$. When there is no strong correlation between two variables, $\rho$ is either equal or close to 0. When one variable monotonically increases with another, $\rho = 1$, and when one monotonically decreases, $\rho = -1$.

### 2.2. Grey Wolf Optimization Algorithm

The Grey Wolf Optimization algorithm (GWO) was proposed by Mirjalili and others in 2014. The basic principle is to imitate the population system of grey wolves, and divide them into four levels: α wolves, β wolves, δ wolves, and ω wolves. The above four levels correspond to the optimal solution, the optimal solution, the suboptimal solution, and the candidate solution of the optimization problem, respectively [24]. The optimization

process of GWO is guided by α, β, and δ. After judging the prey position as the optimal solution, it guides ω around the prey and finds the optimal value through continuous iteration. The process of the Grey Wolf Optimization algorithm can be divided into three stages: encirclement, pursuit, and attack. The specific steps are as follows:

The hunting process, the gray wolf rounding up prey behavior is defined as follows:

$$\vec{D} = \left| \vec{C} \cdot \vec{X}_P(t) - \vec{X}(t) \right| \tag{2}$$

$$\vec{X}(t+1) = \vec{X}_p(t) - \vec{A} \cdot \vec{D} \tag{3}$$

Equations (2) and (3) represent the distance between the wolf and the prey and the update distance of the wolf position, respectively. Specifically, they are the position vector of the grey wolf (potential solution vector) and the position vector of the prey (optimal global solution). $t$ is the wolf pack position iteration Times, and both are coefficient vectors. Calculated as follows:

$$\vec{A} = 2\vec{\alpha} \cdot \vec{r}_1 - \vec{\alpha} \tag{4}$$

$$\vec{C} = 2 \cdot \vec{r}_2 \tag{5}$$

The convergence factor, which linearly decreases from 2 to 0 as iterations, is a random vector [0,1].

In the optimization problem decision space, to better search for the position of the prey, it is usually guided by α, β, and δ. At the same time, other grey wolf individuals (including ω) update their positions according to the role of the optimal grey wolf individual. They gradually approach the prey. The mechanism of individual wolves tracking the location of their prey is shown in Figure 1.

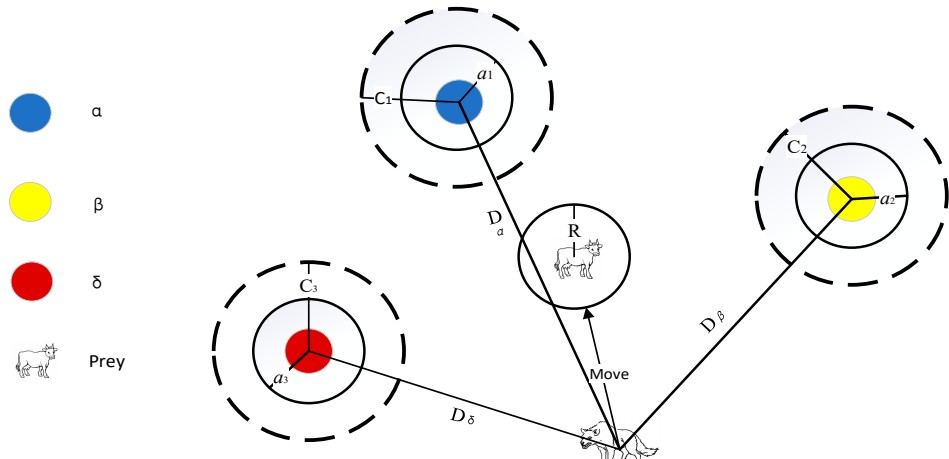

**Figure 1.** Grey wolf location update.

The grey wolf individual tracking prey position model is as follows:

$$\begin{cases} \vec{D}_\alpha = \left| \vec{C}_1 \cdot \vec{X}_\alpha - \vec{X} \right| \\ \vec{D}_\beta = \left| \vec{C}_1 \cdot \vec{X}_\beta - \vec{X} \right| \\ \vec{D}_\delta = \left| \vec{C}_1 \cdot \vec{X}_\delta - \vec{X} \right| \end{cases} \tag{6}$$

$\overrightarrow{D}_a$, $\overrightarrow{D}_{\beta,}$ and $\overrightarrow{D}_\delta$ represent the distance between $\alpha$, $\beta$, and $\delta$, and other individuals, respectively. $\overrightarrow{X}_a$, $\overrightarrow{X}_{\beta,}$ and $\overrightarrow{X}_\delta$ represent the current positions of $\alpha$, $\beta$, and $\delta$, respectively, and are random vectors, which are the existing positions of grey wolves.

$$\begin{cases} \overrightarrow{X}_1 = \overrightarrow{X}_\alpha - A_1 \cdot \overrightarrow{D}_\alpha \\ \overrightarrow{X}_2 = \overrightarrow{X}_\beta - A_1 \cdot \overrightarrow{D}_\beta \\ \overrightarrow{X}_3 = \overrightarrow{X}_\delta - A_1 \cdot \overrightarrow{D}_\delta \end{cases} \tag{7}$$

$$\overrightarrow{X}(t+1) = \frac{\overrightarrow{X}_1 + \overrightarrow{X}_2 + \overrightarrow{X}_3}{3} \tag{8}$$

Attacking means catching the prey, that is, finding the optimal solution. To simulate approaching the prey, it is mainly realized by the gradual decrease in the ground value in Equation (5). When the value linearly decreases from 2 to 0, the corresponding value changes in the interval $[-2\alpha, 2\alpha]$. At that time, the wolves can focus their attacks on their prey. At this time, the wolves will disperse from the position of the prey and enter the process of finding other local optimal solutions. This makes the grey wolf algorithm fall into the optimal local solution.

### 2.3. Levi Flight Improved Grey Wolf Optimization Algorithm

In the Grey Wolf Optimization algorithm, the position of $\alpha$ represents the optimal solution. The grey wolves in the later population all approached the $\alpha$ wolves, resulting in the loss of population diversity. Thus, they fall into local convergence and premature convergence. Aiming to address these shortcomings, this paper uses Levi flight to perform a global search on the group's grey wolf individual $\alpha$ wolves. Levy flight is a random walk, which can expand the search range. Its flight step size satisfies a stable heavy-tailed distribution [19]. The new generation $\alpha$ wolf calculation formula improved by Levi's flight is as follows:

$$\overrightarrow{X}_\alpha(t+1) = \overrightarrow{X}_\alpha(t) - \alpha \oplus levy(\beta) \tag{9}$$

$$Levy(\beta) = 0.01 \frac{\mu}{|v|^{1/\beta}} \left( \overrightarrow{X}_\alpha(t) - \overrightarrow{X}_{\alpha best} \right) \tag{10}$$

$$u = N\left(0, \sigma_u^2\right); v = N\left(0, \sigma_v^2\right) \tag{11}$$

$$\sigma_u = \left\{ \frac{\Gamma(1+\beta)\sin\left(\frac{\pi\beta}{2}\right)}{\Gamma\left[\left(\frac{1+\beta}{2}\right)\right]\beta 2^{(\beta-1)/2}} \right\}^{1/\beta} \tag{12}$$
$$\sigma_v = 1$$

The parameter $\beta$ is a random number of [0,3].

### 2.4. Debris Flow Outburst Scale Prediction Model Based on IGWO-SVM

Support vector machine shows great advantages in solving small sample, nonlin-ear, and high-dimensional identification. Therefore, this paper chooses this model as the basic prediction model. The core parameters of the SVM model are the penalty factor ($c$) and the kernel function parameter ($g$). Using default parameters may lead to overfitting or underfitting issues. Therefore, the proposed IGWO algorithm is employed to optimize these two parameters for SVM, resulting in a debris flow scale prediction model based on IGWO-SVM. The process of the IGWO-SVM debris flow outflow model is shown in Figure 2.

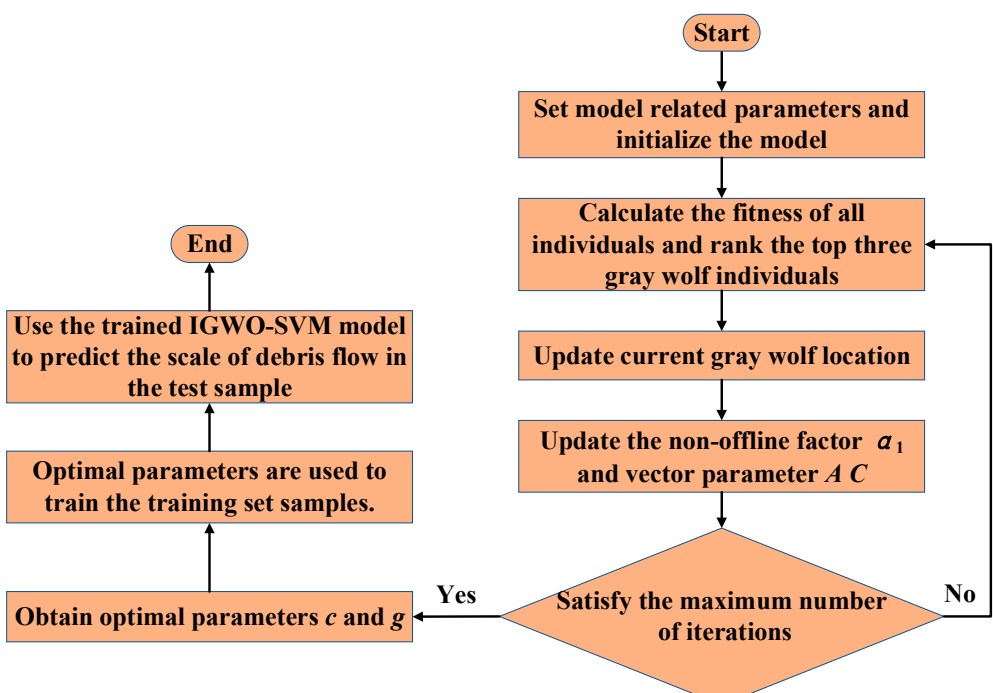

**Figure 2.** IGWO-SVM Debris Flow Scale Prediction Modelling Process.

Specific steps are as follows:

- Step 1: Set the parameters of IGWO and SVM algorithms and initialize the grey wolf population.
- Step 2: Use the minimum recognition error rate of SVM for training set samples as the fitness function, calculate the fitness of all individuals in the population, and sort according to the size of the fitness value to determine the top three grey wolves.
- Step 3: Update the current position of the grey wolf individual according to Equations (10) and (12).
- Step 4: Update the value of the nonlinear convergence factor a according to Equation (13), and update the parameter vectors A and C according to Equations (8) and (9).
- Step 5: Introduce the Levy flight strategy to the grey wolf population according to Equation (14) and adjust the position of the grey wolf.
- Step 6: Determine whether the algorithm has reached the maximum number of iterations. If it is reached, the position of wolf a is returned as the optimal parameter value of SVM. If it is not reached, skip to step 2.
- Step 7: Use the optimal penalty factor c and kernel function parameter g to train and learn the training set samples to obtain the IGWO-SVM fault diagnosis model.
- Step 8: Input the test set samples into the trained IGWO-SVM model to predict the scale of debris flow outburst.

Firstly, Spearman correlation analysis is utilized to select input factors and eliminate poorly correlated factors to enhance model accuracy. After correlation analysis, 50 data are randomly used as the training set, and the remaining data are used for the prediction set. Subsequently, Levy flights are employed to optimize the GWO algorithm, resulting in the development of the Improved GWO algorithm (IGWO). The IGWO was then utilized to optimize the SVM algorithm to obtain the final prediction model for mudslide volume. The final model incorporates the training set for training, followed by validation with the prediction set. Figure 3 illustrates the complete workflow.

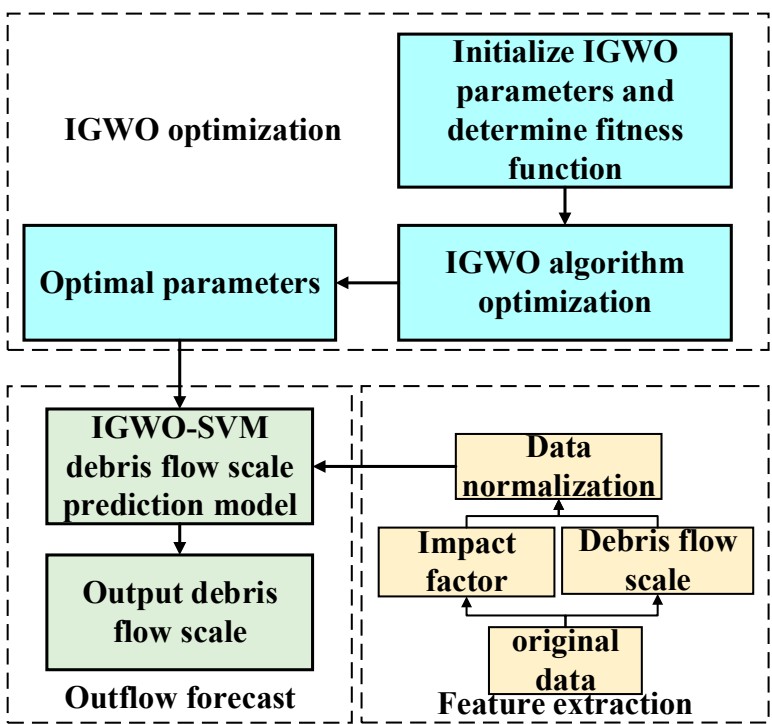

**Figure 3.** Debris flow outburst scale based on IGWO-SVM.

### 2.5. Back Propagation Neural Network

Back Propagation Neural Network (BPNN) [25,26], the most extensively applied and sophisticated neural network model, sees widespread use across various civil engineering domains. The network comprises an input layer, an implicit layer, and an output layer. Weight values between the layers are obtained via signal forward propagation and error backpropagation, culminating in the construction of the prediction model. BPNN serves as the comparison model in this study, facilitating performance comparisons with SVM, GWO-SVM, and IGWO-SVM models.

In this paper, the minimum error for training has been established as 0.001, with the number of training sessions set to 1000 and the learning rate set to 0.1.

## 3. Application Research and Method Comparison

### 3.1. Introduction to Geology and Hydrology of Study Area

Beichuan County is predominantly hilly, featuring high terrain in the western part of the north, moderate slopes in the central region, and lower mountains in the eastern portion of the southern area. The topography is primarily a result of erosion and dissolution. The county is located on the southeastern margin of the tectonic erosion feature known as Zhongshan. At the same time, it acts as the junction of the mountains in the geological area of Longmen Mountain. The range extends towards the northeast in a southwesterly direction. The topography of the county exhibits substantial variation, with high terrain in the northwest and low terrain in the southeast. The difference in altitude exceeds 1000 m. Gully valley slopes usually exceed 25 degrees, while some slope angles reach 40 to 50 degrees or even steeper. In the study area, the Paleozoic eras of Cambrian, Silurian, Devonian, and Carboniferous, along with the loose stacked strata of the Cenozoic era of Quaternary, are present. Figure 4 shows the geology of Beichuan County.

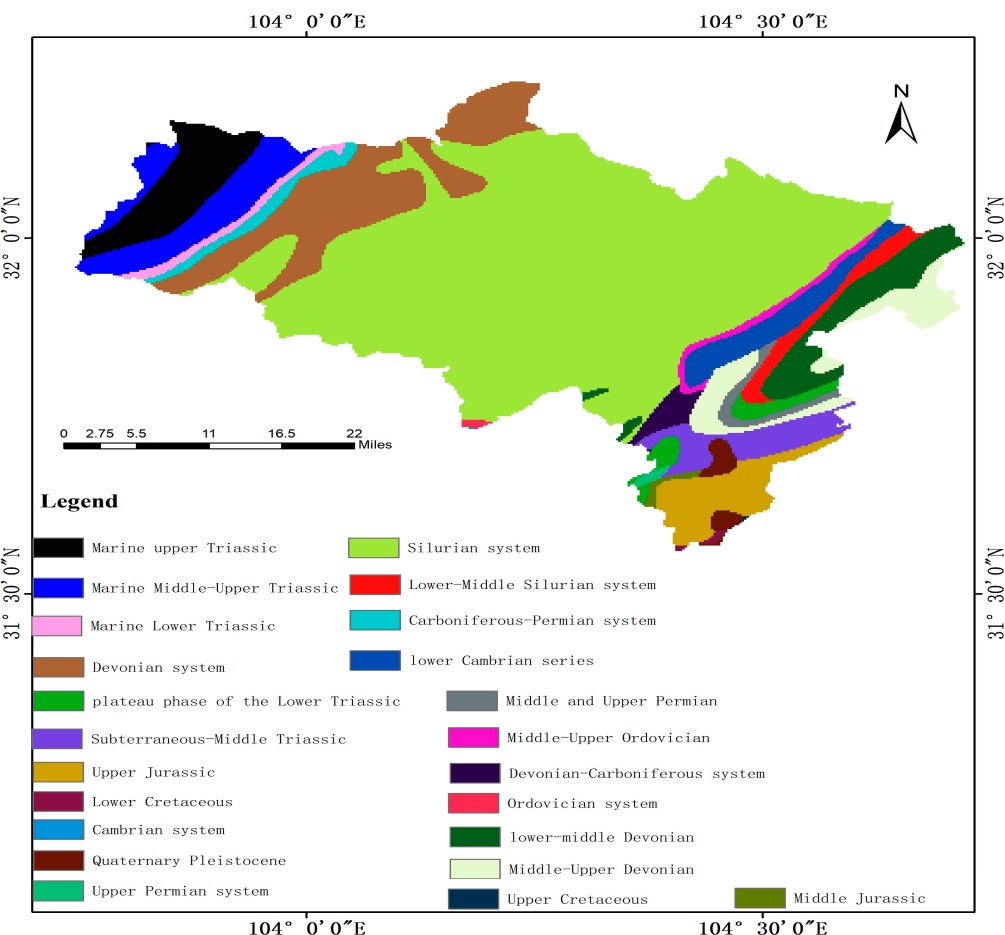

**Figure 4.** Geological map of Beichuan County.

Beichuan County boasts plentiful water resources, sourced mainly from the Wai, Subao, Pingtong, and Duba rivers; the Waijiang River takes precedence in Beichuan County, serving as a premier tributary of the Fuling River. It originates in the northwest mountainous region of the county, flows through the area, exits via the southeast corner, and eventually empties into the Fuling River. The Waijiang River has a length of 47.9 kilometers and flows through Beichuan Qiang Autonomous County, with a watershed area covering 455.80 square kilometers. It has a natural drop of 203 m and an average specific drop of 4.2 per thousand. The multi-year average runoff measures 102.7 cubic meters per second, with a total annual average runoff of 3.257 billion cubic meters. Furthermore, the Waijiang River annually transports 4–5 million tons of sand. The study area boasts ample groundwater resources.

The hydrogeological conditions in Beichuan County prove intricate, influenced by the stratigraphic lithology, topography, and tectonics present in the region. The hydrogeological conditions in Beichuan County are quite intricate. Groundwater in the region is classified into loose rock-type pore water, clastic rock-type pore and fissure water, carbonate rock-type fissure cave water, and bedrock fissure water. The storage patterns of the distinct groundwater types vary, influenced by topography, lithology, tectonic part, and the spatial combination of tectonics. The pore water of loose rock is mainly deposited in the sand, pebble, and gravel layers of the fourth system. It is mainly distributed in the floodplains and low terraces of Waijiang River, Baicao River, Qingpian River, and its tributaries. The water level in the floodplain or first-class terrace of Waijiang, Baicao River, and Qingpian River ranges from 1–6 m deep, indicating a high-water content. Pore and fissure water contained in clastic rock are preserved in the fine sandstone, quartz sandstone, mud shale, dark grey and grey-green fine sandstone, and muddy sandstone within the Qingping Formation of

the Lower Cambrian System and the Lower Devonian System. The argillaceous sandstone of the formation is relatively aquifuge, and the formation is typically a thick to extremely thick layer. The formation fissures are not well developed, resulting in less groundwater. Carbonate fracture cave water exists in the eastern part of the work area. The water-bearing zone is mainly distributed in the northeast direction. The fissure caves of Middle Devonian, Carboniferous, and Permian carbonate rocks are enriched in the two flanks of the dorsal incline and the core of the dorsal incline. The area is characterized by surface dissolution depressions, drop holes, funnels, caves, and even dark rivers. Bedrock fissure water includes tectonic fissure water and metamorphic fissure water. It is widely distributed in the western part of the work area and occurs in the Silurian Maoxian Group (Smx) strata. Tectonic fissure water is situated in high mountains, resulting in limited visible spring outcrops on the surface. Objective evaluation indicates that fewer outflows are present due to the location of the water source.

*3.2. Parameter Selection*

After the 5.12 Wenchuan earthquake, most of the loose sediments on the hillside produced many loose materials. These loose sediments provide favorable conditions for the development and occurrence of debris flow. After the torrential rain on 24 September, 214 geological disasters occurred, including 72 mudslides. The distribution is shown in Figure 5. It has brought great challenges to the resettlement and reconstruction work of residents in the disaster area. This article is based on a survey of debris flow information in 72 valleys in Beichuan County, Sichuan Province [27]. In Table 1, five factors that can comprehensively reflect the material and energy sources of debris flows, namely basin area, main channel length, basin relative relief, basin relative relief, and bed shifting ratio, are selected as influencing factors of the scale of debris flow outburst.

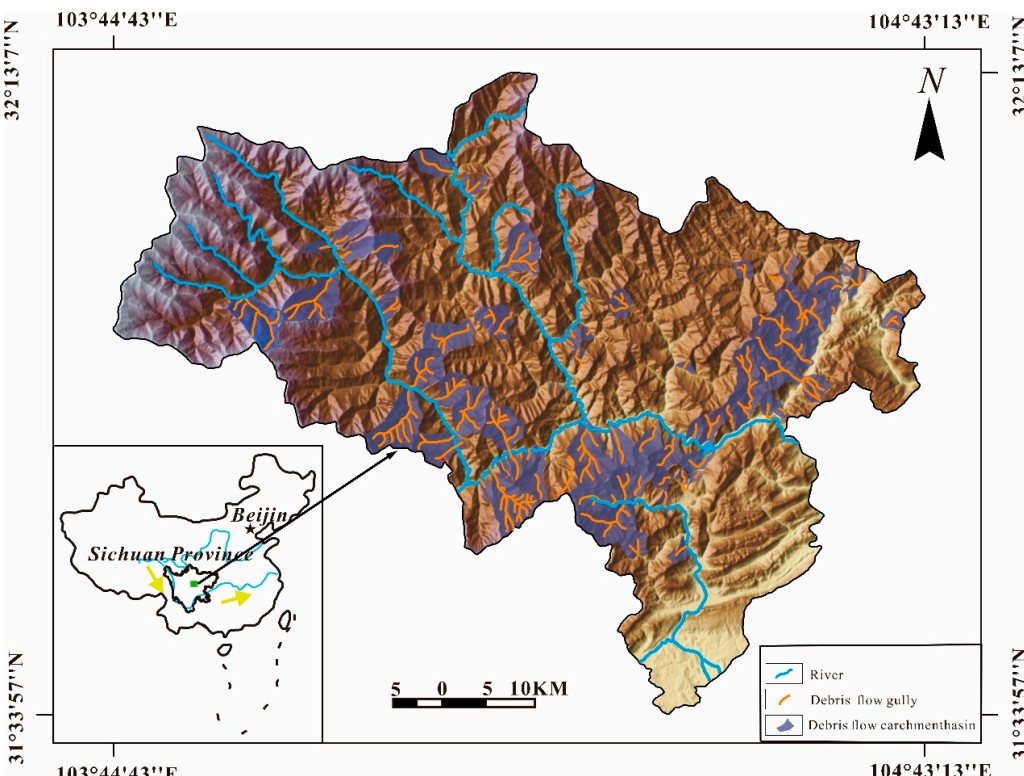

**Figure 5.** Distribution of debris flow in Beichuan County.

**Table 1.** The Basic Data Statistics Table of 72 Debris Flows.

| | The Basic Data Statistics Table of 72 Debris Flows. | | | | | |
|---|---|---|---|---|---|---|
| **Samples** | **Loose Source Material Reserves ($10^3$ m$^3$)** | **Basin Area (km$^2$)** | **Drainage Density (km$^{-1}$)** | **Basin Relative Relief (km)** | **Shifting Bed Proportion (%)** | **Main Channel Length (km)** |
| Chaimazigou#1 | 0.04 | 2.5 | 8.24 | 1.6 | 0.48 | 2.06 |
| Shuxuegou | 39.04 | 13.9 | 2.90 | 1.4 | 0.50 | 4.03 |
| Yingtaogou#1 | 43.65 | 10.3 | 3.78 | 1.4 | 0.72 | 3.89 |
| Miaobagou | 728.20 | 7.8 | 5.26 | 1.46 | 0.85 | 4.10 |
| Jinlongcun | 79.50 | 4.5 | 7.44 | 0.98 | 0.64 | 3.35 |
| Hualingou | 385.95 | 12.2 | 4.82 | 1.36 | 0.85 | 5.88 |
| Wangjiashangou | 104.50 | 1.8 | 7.67 | 1.04 | 0.86 | 1.38 |
| Xinzhigou#1 | 240.45 | 10 | 5.39 | 1.56 | 0.76 | 5.39 |
| Chenjiabaogou | 50.20 | 1.9 | 7.16 | 1.1 | 0.48 | 1.36 |
| Pijialianggou | 2.40 | 2.4 | 7.42 | 1.14 | 0.23 | 1.78 |
| Xishanpogou | 1500 | 1.6 | 20.75 | 1.12 | 0.61 | 3.32 |
| Renjiapinggou | 242 | 0.5 | 14.6 | 0.46 | 0.84 | 0.73 |
| Mofanggou | 160.70 | 0.8 | 13.63 | 0.66 | 0.72 | 1.09 |
| Miaobagou | 6.60 | 7.5 | 3.81 | 1.38 | 0.39 | 2.86 |
| Piankoxianggou#2 | 4.80 | 4.6 | 4.33 | 0.86 | 0.54 | 1.99 |
| Xinzhigou#2 | 73.20 | 21.8 | 3.59 | 2.04 | 0.42 | 7.82 |
| Honglingou | 2.85 | 5.7 | 5.35 | 1.92 | 0.37 | 3.05 |
| Chaimazigou#2 | 14.70 | 6.8 | 3.75 | 1.8 | 0.40 | 2.55 |
| Qinglingou | 109.30 | 23.2 | 3.23 | 2.3 | 0.61 | 7.49 |
| Baishuihegou | 35 | 10.6 | 4.01 | 1.68 | 0.47 | 4.25 |
| Piankoxianggou#3 | 160.34 | 16 | 3.68125 | 1.04 | 0.51 | 5.89 |
| Subaohegou | 60 | 3.5 | 6.43 | 1.24 | 0.65 | 2.25 |
| Shuligou | 70.60 | 0.7 | 20.43 | 0.96 | 0.61 | 1.43 |
| Xinigou | 40.53 | 0.7 | 19.43 | 1 | 0.81 | 1.36 |
| Tianbaigou | 163.32 | 18.7 | 3.16 | 1.68 | 0.76 | 5.91 |
| Piankoxianggou | 0.89 | 0.9 | 15.11 | 0.72 | 0.43 | 1.36 |
| Lijiawangou | 60 | 1.2 | 12.08 | 0.86 | 0.41 | 1.45 |
| Kaipingzhigou | 26.20 | 1 | 13.20 | 0.6 | 0.62 | 1.32 |
| Yuxuegou | 1016.40 | 0.8 | 14.38 | 0.88 | 0.86 | 1.15 |
| Xiatongbaogou | 1967.90 | 15.7 | 3.80 | 1.22 | 0.84 | 5.97 |
| Sibapinggou | 378.24 | 21.4 | 3.47 | 1.5 | 0.76 | 7.42 |
| Zhibeigou | 199 | 8.7 | 3.25 | 1.36 | 0.60 | 2.83 |
| Yangliucun | 101.63 | 9.9 | 4.64 | 1.7 | 0.58 | 4.59 |
| Yanghuziwangou | 40.20 | 1.2 | 12.08 | 0.82 | 0.81 | 1.45 |
| Zhifanggou | 74 | 1.1 | 9.55 | 0.75 | 0.69 | 1.05 |
| Yingtaogou#2 | 119.30 | 17.6 | 4.33 | 1.66 | 0.56 | 7.62 |
| Sunjiagou | 15.55 | 2.7 | 10.70 | 1.22 | 0.45 | 2.89 |
| Chayuanlianggou | 54 | 2.6 | 12.04 | 1.26 | 0.41 | 3.13 |
| Hanjiashangou | 67.44 | 0.8 | 15.25 | 0.82 | 0.82 | 1.22 |
| Baiguoshugou | 107.30 | 0.6 | 16.50 | 0.67 | 0.73 | 0.99 |
| Weigou | 33.54 | 2.2 | 9.50 | 0.74 | 0.57 | 2.09 |
| Weigou#2 | 106.50 | 0.3 | 22.00 | 0.52 | 0.76 | 0.66 |
| Madiwangou | 3.36 | 0.7 | 29.86 | 0.55 | 0.47 | 2.09 |
| Huangjiawangou | 4.13 | 2.8 | 8.39 | 1 | 0.47 | 2.35 |
| Jingzhuyuangou | 51.80 | 1.1 | 9.00 | 0.59 | 0.46 | 0.99 |
| Jiangjiagou | 12.14 | 0.5 | 23.00 | 0.92 | 0.52 | 1.15 |
| Maoershi | 10.80 | 1.4 | 7.57 | 0.98 | 0.47 | 1.06 |
| Subaogou | 507 | 1.1 | 10.45 | 0.58 | 0.79 | 1.15 |
| Liujiagou | 120.08 | 1.8 | 7.50 | 1.04 | 0.89 | 1.35 |
| Daokaimengou | 15.98 | 3.1 | 8.19 | 0.84 | 0.51 | 2.54 |
| Qingtangwangou | 30 | 3.5 | 5.14 | 0.82 | 0.75 | 1.80 |
| Huangtulianggou | 114 | 24.6 | 3.29 | 1.22 | 0.64 | 8.10 |
| Guanmenzigou | 14.26 | 2.8 | 5.57 | 1.12 | 0.70 | 1.56 |
| Shupinggou | 33 | 4.1 | 8.88 | 1.09 | 0.46 | 3.64 |
| Dengjiacungou | 900.03 | 22.2 | 5.12 | 1.7 | 0.44 | 11.36 |

**Table 1.** *Cont.*

| Samples | Loose Source Material Reserves ($10^3$ m$^3$) | Basin Area (km$^2$) | Drainage Density (km$^{-1}$) | Basin Relative Relief (km) | Shifting Bed Proportion (%) | Main Channel Length (km) |
|---|---|---|---|---|---|---|
| Qushanzhenggou | 210 | 3.6 | 8.67 | 1.2 | 0.96 | 3.12 |
| Guzhubagou | 1000.10 | 7 | 5.74 | 1.22 | 0.87 | 4.02 |
| Wangjiayangou | 485 | 2.5 | 7.88 | 1 | 0.81 | 1.97 |
| Chenjiabagou | 931.24 | 23.1 | 4.28 | 1.2 | 0.66 | 9.88 |
| Tudilianggou | 12.21 | 4 | 6.40 | 1.03 | 0.53 | 2.56 |
| Tudimiaogou | 34.08 | 16 | 3.69 | 1.28 | 0.39 | 5.91 |
| Guaitangou | 0.08 | 11.7 | 4.05 | 1.08 | 0.21 | 4.74 |
| Dapingdigou | 16.80 | 5.4 | 5.59 | 1.46 | 0.40 | 3.02 |
| Xiatongbaogou | 98.50 | 22.7 | 3.37 | 1.86 | 0.76 | 7.66 |
| Chanzipinggou | 67.20 | 2.5 | 5.28 | 1.02 | 0.83 | 1.32 |
| Shangyantaigou | 17.50 | 1.5 | 11.67 | 1.24 | 0.9 | 1.75 |
| Shuangyigou | 93.30 | 2.8 | 9.82 | 1.3 | 0.78 | 2.75 |
| Shilonggou | 50.80 | 7.3 | 5.36 | 1.2 | 0.84 | 3.91 |
| Yangjiawangou | 135.57 | 26.4 | 3.20 | 1.8 | 0.67 | 8.46 |
| Zhaojiawangou | 14.66 | 2.8 | 8.18 | 1.34 | 0.82 | 2.29 |
| Dongxigou | 8.95 | 10.9 | 3.78 | 1.5 | 0.57 | 4.12 |
| Maliuwangou | 97.82 | 17.1 | 3.76 | 1.28 | 0.70 | 6.43 |

### 3.3. Data Presentation and Evaluation

The selected training data source is Wang's thesis on the debris flow in Beichuan County following the fifth. This study analyses the data from 72 debris flow samples following a 2012 earthquake. Each sample included six parameters, including Loose source material reserves, Basin area, Drainage density, Basin relative relief, Shifting bed proportion, and Main channel length. These variables are the most common factors impacting the scale of mudslide outflow, the output parameter. Results of the parameter statistical analysis can be found in Table 2, and Figure 6 displays the frequency distribution graphs for each parameter. Where 'n' represents the frequency number, indicating the count of samples in each sub-interval of the variable, the frequency ratio of each sub-interval to the total sub-frequency of the variable is referred to as the frequency. Furthermore, 'F' denotes the cumulative frequency achieved through the incremental addition of frequencies of each sub-interval.

**Table 2.** Parameter statistics.

| Data Type | Loose Source Material Reserves ($10^3$ m$^3$) | Basin Area (km$^2$) | Drainage Density (km$^{-1}$) | Basin Relative Relief (km) | Shifting Bed Proportion (%) | Main Channel Length (km) | Debris Flow Scale |
|---|---|---|---|---|---|---|---|
| minimum value | 0.04 | 0.3 | 10.68 | 0.46 | 0.21 | 0.66 | 6.3 |
| maximum value | 1966.9 | 26.4 | 44.06 | 2.3 | 0.96 | 11.36 | 152.83 |
| average value | 195.93 | 7.10 | 22.18 | 1.18 | 0.63 | 3.40 | 62.85 |

Considering each factor and the debris flow scale, respectively, the correlation results are shown in Table 3 and Figure 7. The single factor correlation analysis shows that the basin area, the relative channel of the basin, and the length of the main channel have a high correlation with the scale of debris flow. Therefore, these three are selected as the influencing factors of the debris flow scale to construct a prediction model of the debris flow scale.

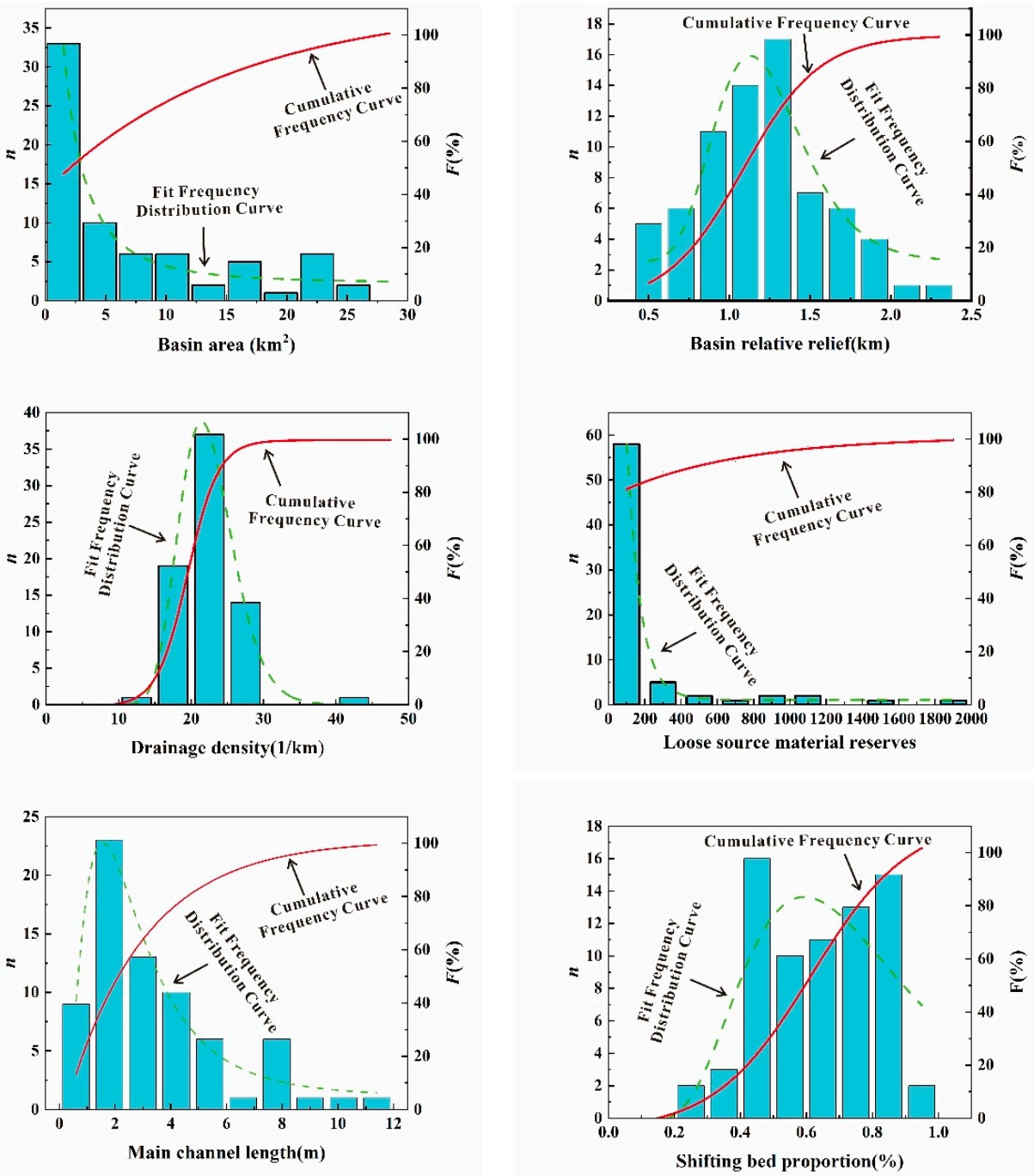

**Figure 6.** Frequency distribution.

**Table 3.** Correlation Analysis.

| Correlation Analysis | |
|---|---|
| **Correlation Factor** | **Debris Flow Scale ($10^3$ m$^3$)** |
| Basin area/km$^2$ | 0.920 ** |
| Drainage density/1/km | 0.136 |
| Basin relative relief/km | 0.778 ** |
| Shifting bed proportion/% | −0.154 |
| Main channel length/km | 0.766 ** |

Note: ** $p < 0.01$.

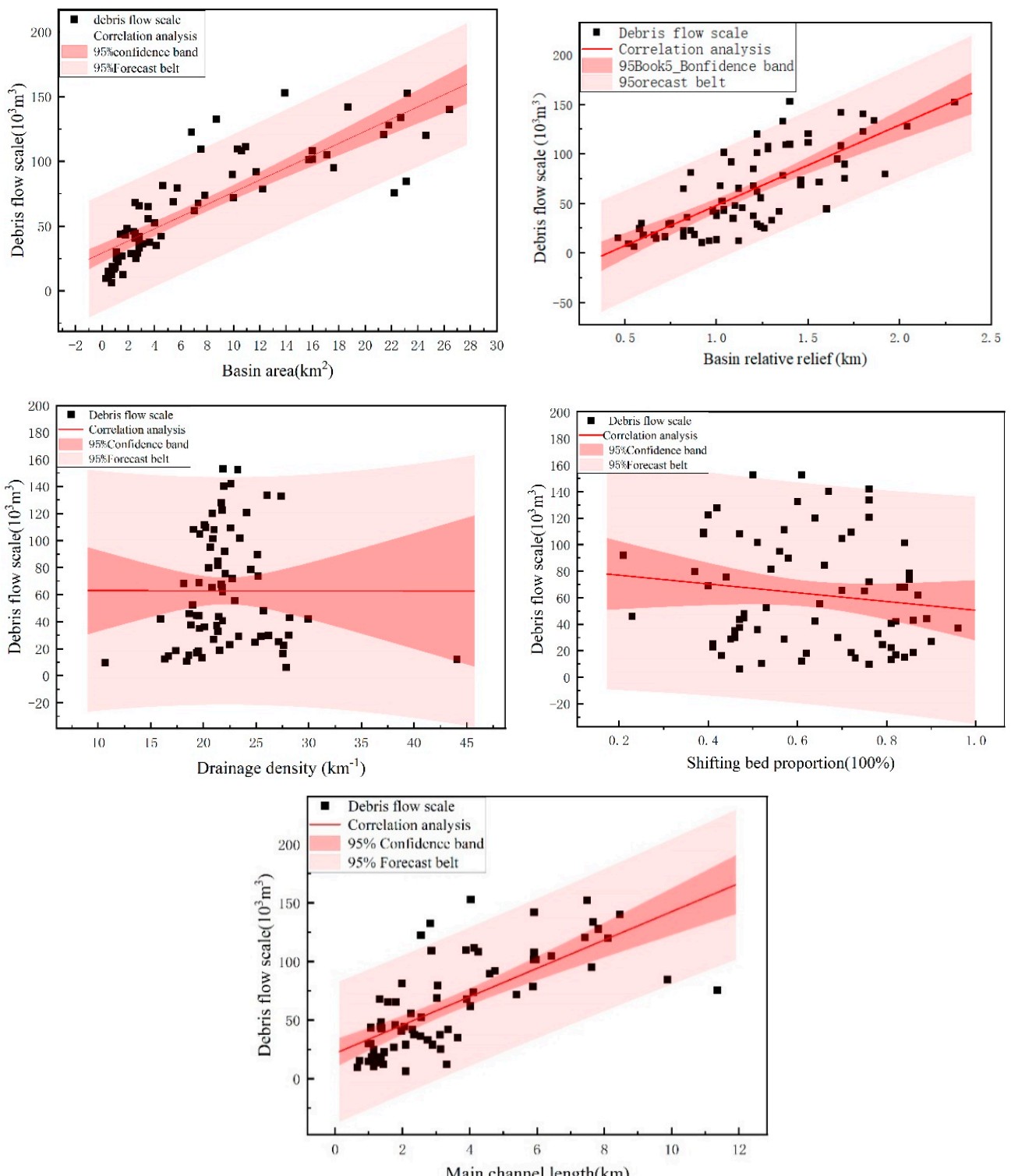

**Figure 7.** Correlation analysis.

### 3.4. Forecast of the Debris Flow Scale

Fifty debris flow data are randomly selected for model training, and the remaining data are used as prediction samples to test the prediction effect of debris flow scale. The results are shown in Figure 8. It can be seen from Figure 8 that the training and prediction of the model have good accuracy.

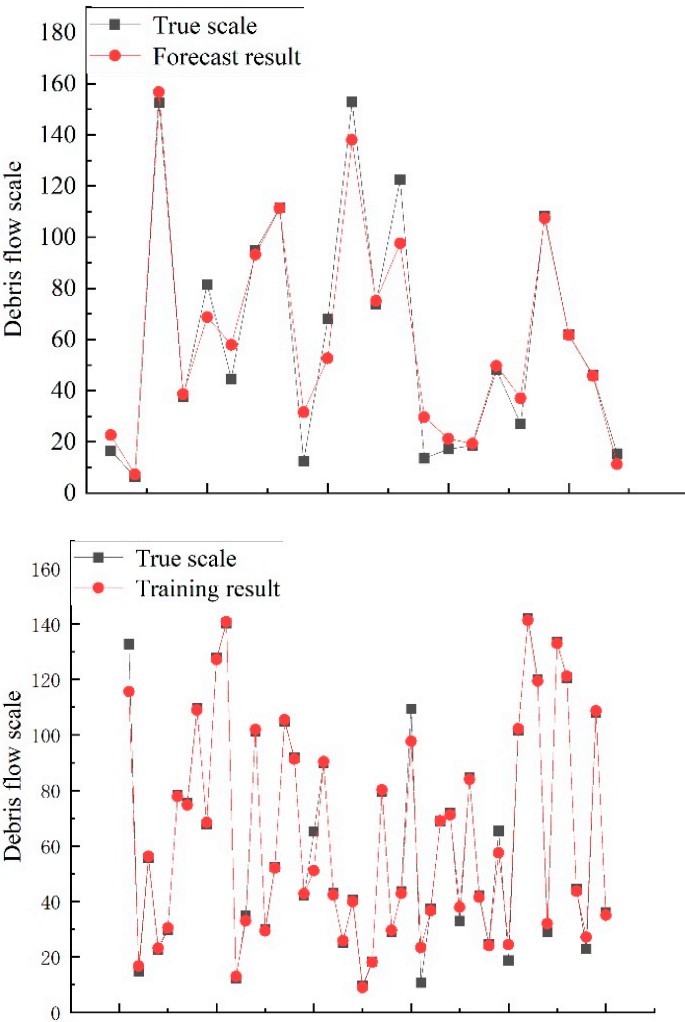

**Figure 8.** Comparison of training and prediction.

*3.5. Model Performance Evaluation*

To further evaluate the predictive performance of the method, this paper selects two traditional fitting methods (linear fitting and power function fitting) and the three intelligent algorithms (SVM, GWO-SVM, and BPNN) as comparisons.

3.5.1. Linear Regression Fitting

The basin area, the relative height difference, and the main channel length are used as independent variables, and the debris flow scale is used as the dependent variable. Linear regression is used. The results are as follows:

$$\begin{aligned} V = 14.818 + 10.334 * S \\ + 39.329 * H - 21.377 * L \end{aligned} \tag{13}$$

The $R^2$ = 0.904 is pretty good in terms of model accuracy alone. However, in this result, the relationship between the length of the main gully and the scale of the debris flow is a negative growth relationship. That is, the longer the length of the main ditch, the smaller the scale of the debris flow. This contradicts the results of correlation analysis. After testing the model, the results are shown in Table 4. The VIF of the basin area and the length of the main river channel are 11.354 and 11.396, respectively, both of which are greater than 10. This shows that there is a more obvious collinearity relationship. This is the main reason for the negative main groove length in the line fitting.

**Table 4.** Linear Regression Analysis Results.

| Linear Regression Analysis Results | | | | | | | | |
|---|---|---|---|---|---|---|---|---|
| Unstandardized Coefficients | | Standardized Coefficient | $t$ | $p$ | VIF | $R^2$ | Adjust $R^2$ | $F$ |
| **B** | **Standard Error** | **Beta** | | | | | | |
| constant | 14.818 | 7.171 | - | 2.066 | 0.044 * | - | | |
| Basin area | 10.334 | 1.035 | 1.538 | 9.988 | 0.000 ** | 11.354 | | |
| Basin relative relief | 39.329 | 7.251 | 0.385 | 5.424 | 0.000 ** | 2.414 | 0.904 | 0.898 | F (3,46) = 144.282 $p$ = 0.000 |
| Main channel length | −21.377 | 3.322 | −0.993 | −6.436 | 0.000 ** | 11.396 | | |

Note: Dependent variable: debris flow scale; D-W: 2.149; * $p < 0.05$, ** $p < 0.01$.

### 3.5.2. Power Function Fitting

When using the power function fitting, the three correlation factors are artificially taken as positive values to obtain the correct correlation relationship. The least-square regression is used to fit the parameters to be sought. The relevant results are as follows:

$$V = a \times S^b \times H^c \times L^d \tag{14}$$

In the above equation $a = 29.275040548$; $b = 0.416174002$; $c = 0.382748483$; and $d = 0.000000029$.

The results show that $R^2 = 0.823$, and the power function can predict the scale of debris flow. However, it is a pity that this formula is easy to mislead the analysis of debris flow impact factors. Because it is not difficult to conclude from the fitting that the length of the main ditch is not critical to the scale of the debris flow. However, the correlation analysis shows that the length of the main gully is highly correlated with the scale of debris flow.

It can be seen from the above two traditional fitting methods that these methods have good accuracy in fitting the debris flow scale. However, these methods often lead to misunderstandings about the factors determining the magnitude of debris flows. These methods make it difficult to find the key factors. To sum up, compared with the intelligent algorithm, the traditional fitting method that can intuitively reflect the influence factors of the debris flow scale does not seem to have any advantages. In summary, this paper is more inclined to use correlation analysis to determine the main influencing factors of debris flow scale, and then build a debris flow scale prediction model through support vector machine.

### 3.5.3. Comparison with Other Common Optimization Algorithms

To highlight the advantages of this method, this paper selects three intelligent algorithms for comparison. And the comparison results are shown in Figure 8.

In general, the four models can predict the debris flow scale, and the effect is good, but the overall IGWO-SVR is the closest to the actual value of the debris flow scale. To evaluate the impact of the prediction model more intuitively for the debris flow scale, this paper will evaluate the prediction model from accuracy and efficiency. The prediction error distribution of the BPNN prediction model is more discrete, and the distribution range of prediction error is more significant than that of SVR and its improved model. This shows that the prediction effect of the BPNN model is poor. Compared with the SVR and GWO-SVR, the IGWO-SVR error distribution is more inferior. It is concentrated near zero, and the error range is lower than the other three methods, which has better stability.

To analyze the overall performance of the prediction model, this paper selects root mean square error (RMSE), average absolute error (MAE), and coefficient of determination ($R^2$) to evaluate the above four prediction models [28].

$$\text{RMSE} = \sqrt{\frac{1}{n}\sum_{i=1}^{n}(\hat{y}_i - y_i)^2} \tag{15}$$

$$\text{MAE} = \frac{1}{n}\sum_{i=1}^{n}|\hat{y}_i - y_i| \tag{16}$$

$$R^2 = 1 - \frac{\sum(\hat{y}_i - y_i)^2}{\sum(\hat{y}_i - y_i)^2} \tag{17}$$

In Equations (15)–(17): $n$ is the size of the sample, $i$ is the $i$-th data sample among n samples, $\hat{y}_i$ is the predicted debris flow scale, and $y_i$ is the accurate debris flow scale. The results are shown in Table 5.

**Table 5.** Prediction Error Analysis of Different Prediction Models.

| Prediction Error Analysis of Different Prediction Models | | | |
|---|---|---|---|
| **Name** | **RMSE** | **MAE** | $R^2$ |
| IGOW-SVR | 7.75 | 7.0 | 0.95 |
| GOW-SVR | 7.80 | 7.6 | 0.94 |
| SVR | 10.99 | 8.79 | 0.92 |
| BPNN | 13.70 | 14.47 | 0.83 |

It can be seen from Table 5 that all four methods can be used as a prediction model for the debris flow scale. The IGWO-SVR model has RMSE = 7.75, MAE = 7.00, $R^2$ = 0.95, which is better than the other three models. It is worth noting that the prediction accuracy of the BPNN is significantly lower than that of the other three. This is due to the significant data demand for BPNN training, which is not suitable for debris flow scales with a small number of statistical samples.

The running time of each debris flow prediction model is calculated on the Intel(R) Core (TM) i5-9300H CPU 2.40 GHz Win10. The results are shown in Table 6. The running time of IGWO-SVR is 1.7876 s. Compared with BP neural network, SVR, and GWO-SVR, the efficiency is increased by 204.88%, 102.66%, and 29.46%, respectively. It shows that the prediction model proposed in this paper has high efficiency in predicting the debris flow scale and is more conducive to practical engineering applications.

**Table 6.** Prediction Model Consumption Time Comparison.

| Prediction Model Consumption Time Comparison | | | | |
|---|---|---|---|---|
| **SVR** | **BPNN** | **GWO-SVR** | **IGWO-SVR** | **SVR** |
| Time/s | 3.6226 | 5.4500 | 2.3141 | 1.7876 |

## 4. Discussion

*Sobol Method for Sensitivity Analysis*

The Sobol method is a quantitative global sensitivity analysis algorithm based on variance decomposition [29]. This method decomposes the total variance of the objective function into individual parameter variances and multi-parameter interaction variances. It finds wide applications in sensitivity analysis. The results of first-order sensitivity indices and global sensitivity indices are shown in Figure 9.

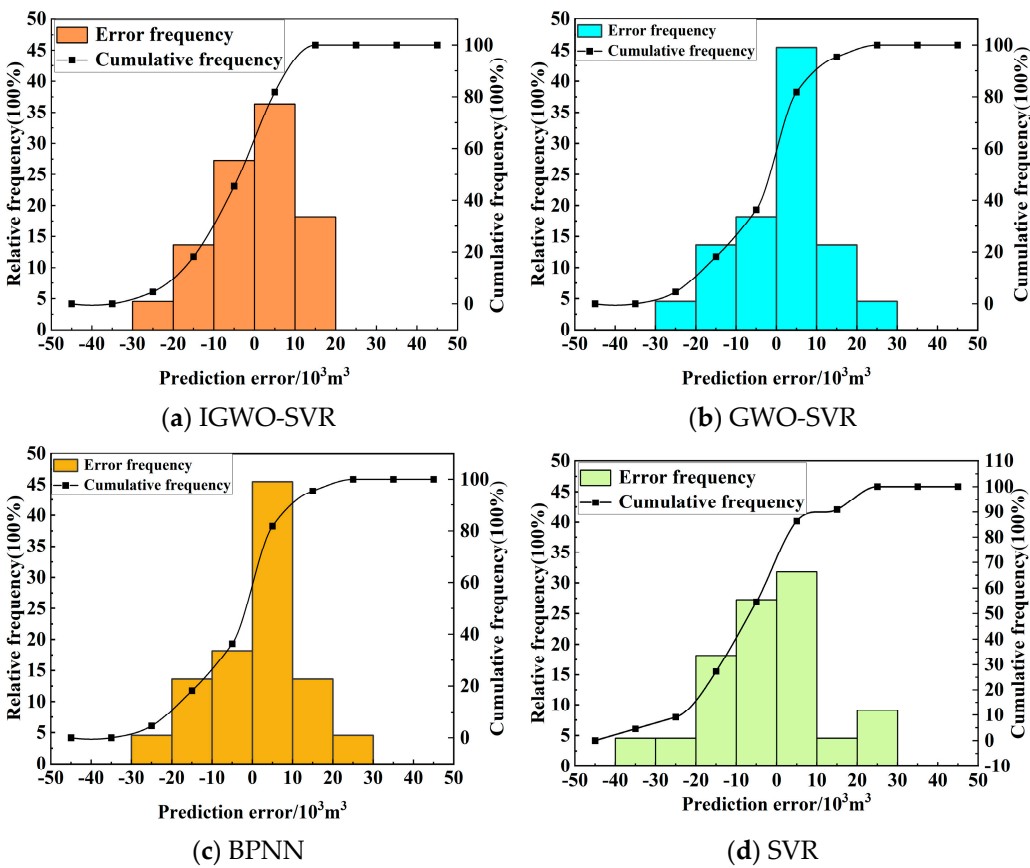

**Figure 9.** Four methods to predict data error frequency distribution.

From Figure 10, we can observe that the global sensitivities of "Basin area", "Basin relative relief", and "Main channel length" with respect to "the debris flow scale" are all greater than 0.2. The "Basin relative relief", which directly represents the potential energy source for debris flows, has the highest first-order sensitivity and global sensitivity indices, which are 0.370 and 0.372, respectively. On the other hand, "Basin area" and "Main channel length", as the most direct indicators of debris flow material sources, are closely related to debris flow discharge. In contrast, the sensitivity indices for "Drainage density" and "Shifting bed proportion" are relatively low, with first-order sensitivity indices of [0.012, 0.013] and global sensitivity indices of [0.080, 0.085]. This is because these two factors typically indirectly influence debris flow material sources, thus affecting the maximum debris flow discharge. It's worth noting that these findings align with the results obtained from the correlation analysis of debris flow influencing factors. Through the sensitivity analysis, the main factors affecting the debris flow scale in Beichuan County are the basin area, the basin relative relief, and the main channel length. Among the above three influencing factors, the relative height difference has an apparent positive relationship with the debris flow scale. This is due to the large amounts of loose deposits produced after the Wenchuan earthquake on 12 May, and the above two influencing factors are the most closely related to loose deposits. Therefore, the debris flow in Beichuan County can be controlled from three aspects: drainage area, relative relief of the drainage basin, and the length of the main river channel. For example, the slope can be cut to reduce the load, reduce the height and slope of the slope, and reduce the risk of deformation and damage of the slope. Or use retaining structures, such as retaining walls, anti-slide piles, etc., to support and reinforce the slope and improve the stability of the slope.

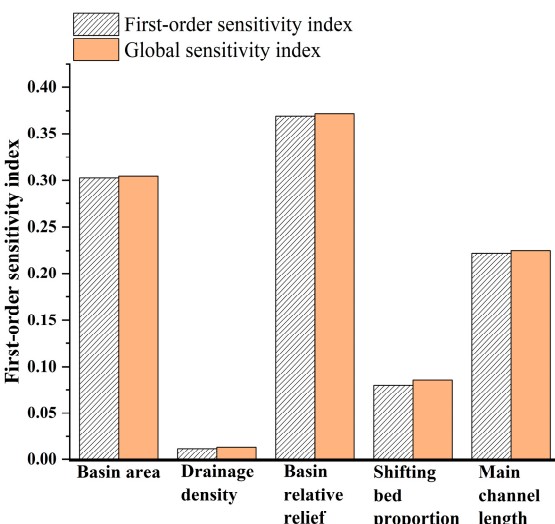

**Figure 10.** Sensitivity index.

It is worth noting that the above three factors are not the main factors for the debris flow in each region. For example, trench length and slope were chosen as dominant factors when Ikeya fitted the debris flow scale in the Pacific region [30]. Therefore, the debris flow scale has a strong regionality. So, it is necessary to carry out correlation analysis when calculating debris flow. At the same time, it is worth noting that although only five slope factors are selected for analysis, the scale of debris flow is affected by various internal and external factors. Lithology, weathering degree of rock, and plant distribution characteristics will all affect the debris flow scale. Therefore, this paper only provides a feasible method to predict the size of debris flow. To further improve the accuracy of the debris flow scale, the number of influencing factors of the debris flow scale should be increased, and quantitative theory should be used to analyze related factors further. And the analysis method in this paper is only for what has occurred or determines the debris flow scale that will happen. Therefore, it should be used in the actual debris flow detection, and it is recommended to be used in conjunction with the prediction of debris flow occurrence.

## 5. Conclusions

This study takes the 72 debris flow in Beichuan County as the research object. Through the correlation analysis, the main influencing factors of the debris flow scale are found, and the improved Grey Wolf Algorithm is used to optimize the support vector regression to train and predict the debris flow scale. By comparing two traditional methods and three machine learning methods, the following conclusions are obtained:

1.  The leading factors of the debris flow scale in Beichuan County are the basin area, the basin relative relief, and the main channel length.
2.  Aiming to address the shortcomings of support vector machines such as slow convergence speed and ease to fall into local extremes, the improved Grey Wolf Algorithm can improve the prediction speed and accuracy of debris flow scale.
3.  With regard to the regional characteristics of Beichuan County, since the three influencing factors of basin area, relative height difference and main ditch length have a greater impact on debris flow, when designing the debris flow prevention and control programme, the focus should be on these three factors for consideration.
4.  The enhanced Grey Wolf Algorithm outlined in this paper lessens the impact of personal opinions and biases on the Debris Flow Scale Prediction process, and the evaluation outcomes give a degree of confidence, thereby offering technological aid for the scientific assessment of Debris Flow danger.
5.  In the next study, it may be considered to add more data sets using numerical simulation to improve the predictive accuracy of the model. However, increasing the data set

will also increase the model run time. Finding a balance between increasing the data set and controlling the model run time is a future direction.

**Author Contributions:** Conceptualization, L.L.; Methodology, Z.Z.; Validation, B.N.; Investigation, H.W.; Data curation, H.L.; Writing—original draft, Z.Z.; Writing—review & editing, D.Z.; Visualization, Y.Q.; Project administration, S.H. All authors have read and agreed to the published version of the manuscript.

**Funding:** This work was supported by the Scientific and Technological Research Program of Chongqing Municipal Education Commission (Grant No. KJZD-M202301205, KJQN202001218, KJQN202301260, KJQN202101206, KJQN202201238), the Research development and application of "big data intelligent prediction and early warning cloud service platform for geological disasters in the Three Gorges Reservoir Area" of Chongqing Municipal Education Commission (Grant No. HZ2021012), the Open fund of Chongqing Three Gorges Reservoir Bank Slope and Engineering Structure Disaster Prevention and Control Engineering Technology Research Center (Grant No. SXAPGC21ZD01), the Science and technology innovation project of Chongqing Wanzhou District Bureau of science and technology (Grant No. wzstc20230303), Nanjing 2022 "Science and Technology Three Gorges" Chongqing Wanzhou District counterpart support project of Chongqing Wanzhou District Bureau of science and technology (Grant No. 2022101S-02), and 2023 Chongqing Postgraduate Research Innovation Project (Grant No. CYS23736).

**Data Availability Statement:** The datasets used and analyzed during the current study are available from the corresponding author upon reasonable request.

**Conflicts of Interest:** The authors declare no conflict of interest.

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
