# Peer review of "Debris Flow Scale Prediction Based on Correlation Analysis and Improved Support Vector Machine"

_water, doi:10.3390/w15234161_

Round 1

Reviewer 1 Report

Comments and Suggestions for Authors

The paper is quite interesting and present an interesting and original method for predicting debris flows occurring. Unfortunately there is no reference to the geological and hydrogeological framework which could make the 72 cases, the paper is referred to, to be comparable. I think the paper needs to be integrated by a sound and detailed description of the geological and hydrogeological properties which characterize the different considered cases and let we evaluated if they can really be statistically dealt with. 

Comments on the Quality of English Language

English language seems to be correct and fluent. It only needs to be revised due to small mistakes (p.e. correct conclusions instead of concusion at the final paragraph.)

Reviewer 2 Report

Comments and Suggestions for Authors

 This paper presents a new computational model for predicting debris flow based on the improved Grey Wolf algorithm for optimising support vector machines, which has a competitive advantage in model representation compared to the traditional fitting approach. However, the following significant concerns require attention:

1. In the manuscript, prior to modelling, it is requested that the authors incorporate a section dedicated to the analysis of debris flow data.

2. The statement in the discussion section of the manuscript is somewhat lengthy and the text in that section needs to be streamlined.

3. The conclusion of the paper inadequately summarises the thesis and requires a comprehensive recapitulation and summarisation.

4. The language presented in the manuscript is suboptimal in terms of academic rigour. It is recommended that the author engage the services of a native English speaker to maximise the clarity and structure of the text.

5. SVR and SVM are mentioned in the manuscript when presenting the model theory, and it is hoped that the authors will standardise the terminology to prevent confusing the reader.

6. Some parts of the manuscript are not clear or ambiguous, and the authors are advised to rephrase and revise them.

Comments on the Quality of English Language

Minor editing of English language required.

Reviewer 3 Report

Comments and Suggestions for Authors

Dear Authors,

Thus study is an interesting application of machine learning for natural hazard assessment. I have the following suggestions to improve the presentation:

1. Lines 28-78: There are many connected sentences that should be separate.

2. An introduction to methodology needs to be given before section 2.1.

3. The algorithms explained in sections 2.1, 2.2, 2.3 and 2.4 are very generic (almost text book style). The rationale for their use with a brief discussion needs to be given. In addition, the selected SVM parameters need to be explained better. I assume that Fig. 4 represents the overall methodology, which is not clear due to its caption and location in the text.

4. Table 1 can be moved to appendix and a statistical summary of this table can be given in this part.

5. Selection of the learning parameters needs to be justified w.r.t. the literature.

6. Section 3.2 need to be expanded with explanations about the correlation results.

7. A comparison with BPNN is given, although this method was never explained in the methodology. Same with the linear regression.

8. A comparative discussion of the results w.r.t. the literature is missing.

Comments on the Quality of English Language

Moderate English revision is needed.

Reviewer 4 Report

Comments and Suggestions for Authors

Dear Authors,

Thank you for your manuscript, predicting debris flow scale is an import work, however in my opinion your work needs more attention before being ready for publication.

Firstly, i think you have to define very well what is your understanding of "debris flow scale" what is included in this term as can leave the reader with doubts, for example, what about the spatial extend, magnitude, how are the incorporated... ?

Abstract - please remove the last sentence as it is very strong statement.

As for the manuscript, i believe your introduction and the literature review should be improved as now it is basically listing of previous work without discussing them, then the introduction finishes with very brief sentence of your proposed work. I believe you have to extend this part with more describing your proposed approach and what is the novelty compared with the previously listed ones. you have to explain what is "gray wolf"
you should extend what are you proposing as methodology what will be the benefits of applying it and the novelty

2. Method and 2.1 Correlation analysis - i believe here you have to insert a general description of your approach and add a workflow scheme

No data section? Please add

Please check the figures as some legend are hard to read

Somehow methodology, results and discussions are merged, please try to separate and define those specific sections.

BPNN is not discussed in the methodology

Table 3, please arrange it properly

Please review your manuscript better with a native speaker or service, as there are many grammatical errors, typos and mixing tenses in it.

Comments on the Quality of English Language

Please review your manuscript better with a native speaker or service, as there are many grammatical errors, typos and mixing tenses in it.

Reviewer 5 Report

Comments and Suggestions for Authors

Dear Editor and Authors,

I have revised the manuscript entitled: Debris Flow Scale Prediction Based on Correlation Analysis and Improved Support Vector Machine. The manuscript tackles an interesting topic, is well written, good methodology, clear objectives and results. I didn’t identify major problems or weaknesses, and in my opinion the manuscript can be considered for publication.

-I suggest to improve the manuscript with further information and explanation regarding the importance of the case study, and therefore, I suggest to move in the section methodology.

-In the “Discussion” section, many parts of the text are methodology not discussion, for example in the accuracy analysis are describe the methods used.

-The discussion section must be better benchmarked with the published literature.

-The conclusions section is expected to have policy recommendations, practical implications and future research.

Round 2

Reviewer 1 Report

Comments and Suggestions for Authors

In the present version the paper can be published, The authors have introduced the geological references I have asked, I only now suggest to introduce also a geological map to better present the geological framework.

Author Response

Response :We have added a geological map of Beichuan County to Figure 5 and described the geology of Beichuan County in section 3.1 in blue type.

Reviewer 2 Report

Comments and Suggestions for Authors

The manuscript can be published.

Comments on the Quality of English Language

Minor revise needed of English language.

Author Response

We have corrected the problems with the grammatical expression of the English language in the article

Reviewer 3 Report

Comments and Suggestions for Authors

Dear Authors,

Thanks for addressing my comments. The manuscript has improved greatly. I suggest to expand the BPNN part by providing the hyperparameters.

Kind regards

Author Response

We have provided the hyperparameters of the BPNN in the blue font section of Section 2.5 in the article

Reviewer 4 Report

Comments and Suggestions for Authors

Thank you for taking into account my comments. I believe now the manuscripti is suitable for publication.

regards.

Author Response

Dear Reviewer:

Thank you for taking time out of your busy schedule to review the manuscript.

Good luck with your work. Have a nice life.